# GaN/Si Heterojunction VDMOS with High Breakdown Voltage and Low Specific On-Resistance

**DOI:** 10.3390/mi14061166

**Published:** 2023-05-31

**Authors:** Xin Yang, Baoxing Duan, Yintang Yang

**Affiliations:** Key Laboratory of the Ministry of Education for Wide Band-Gap Semiconductor Materials and Devices, School of Microelectronics, Xidian University, No. 2 South TaiBai Road, Xi’an 710071, China; bxduan@163.com (B.D.); ytyang@163.com (Y.Y.)

**Keywords:** breakdown voltage, specific on-resistance, heterojunction, interface state, MOSFETs

## Abstract

A novel VDMOS with the GaN/Si heterojunction (GaN/Si VDMOS) is proposed in this letter to optimize the breakdown voltage (*BV*) and the specific on-resistance (*R_on,sp_*) by Breakdown Point Transfer (BPT), which transfers the breakdown point from the high-electric-field region to the low-electric-field region and improves the *BV* compared with conventional Si VDMOS. The results of the TCAD simulation show that the optimized *BV* of the proposed GaN/Si VDMOS increases from 374 V to 2029 V compared with the conventional Si VDMOS with the same drift region length of 20 μm, and the *R_on,sp_* of 17.2 mΩ·cm^2^ is lower than 36.5 mΩ·cm^2^ for the conventional Si VDMOS. Due to the introduction of the GaN/Si heterojunction, the breakdown point is transferred by BPT from the higher-electric-field region with the largest radius of curvature to the low-electric-field region. The interfacial state effects of the GaN/Si are analyzed to guide the fabrication of the GaN/Si heterojunction MOSFETs.

## 1. Introduction

Power semiconductor devices, also known as electronic power devices, are the main object of power electronics research. A vertical double-diffused MOS (VDMOS) was proposed by H.W. Collins in 1979 [1,2]. However, there is a contradictory relationship between the *BV* and *R_on,sp_* of VDMOS, and *R_on,sp_* increases sharply with the increase in *BV*. In order to alleviate this contradictory relationship, several new structures were proposed [3,4,5,6,7,8,9,10,11,12,13,14,15,16,17]. With the advances in the technology, the functions of Si-based power devices are gradually approaching the limits of Si materials. In comparison to silicon, GaN, as a third-generation semiconductor material, possesses characteristics such as a wide bandgap and high critical breakdown electric field, making it the preferred material for high-frequency and high-power devices [8,9,10,11,12]. However, due to the technical immaturity of Si-based GaN devices in terms of source ohmic contacts and gate oxidation processes, their manufacturing poses significant challenges, resulting in difficulties achieving optimal electrode formation, greatly limiting the development and application of GaN devices [8,9].

However, challenges persist regarding the reliability of the gate oxide layer and the implementation of high-quality ohmic contacts in GaN devices. To address this issue, the introduction of heterogeneous junction structures has garnered significant attention [10,18,19,20]. The proposed novel VDMOS device with GaN/Si heterogeneous junction combines the advantages of GaN material’s wide bandgap and high breakdown field strength. Consequently, compared to traditional silicon-based power devices, this structure can achieve a higher breakdown voltage. Additionally, the GaN/Si VDMOS device combines mature Si process technologies to enable the formation of high-quality ohmic contacts. Therefore, in contrast to Si-based GaN devices, this structure allows for the use of mature Si process technologies in gate oxide and ohmic contact, significantly reducing the manufacturing challenges and achieving high-quality gate oxide and ohmic contact [21,22,23,24,25,26].

The GaN/Si VDMOS device optimizes the contradictory relationship between *BV* and *R_on,sp_* through a breakdown point transfer technique (BPT). In traditional Si VDMOS, leakage current accumulates in area A (as shown in Figure 1), which means that the breakdown point of traditional Si VDMOS is located at the bottom of the P-base region with the largest curvature radius and the N-drift region (area A in Figure 1). When the highest electric field at this point reaches the critical breakdown field strength of Si, traditional Si VDMOS experiences breakdown in area A. In the case of GaN/Si VDMOS, the leakage current accumulates in area B (as shown in Figure 1). Although the highest electric field is formed at area A, it does not reach the critical breakdown field strength of the GaN material in GaN/Si VDMOS. Instead, the lower electric field in area B eventually reaches the critical breakdown field strength of Si under a high drain voltage, causing breakdown in area B of GaN/Si VDMOS. This means that the breakdown in SiC/Si VDMOS devices are transferred from area A with the largest curvature radius to area B through the breakdown point transfer technique, similar to RESUEF and REBULF technologies, by reducing the highest electric field to improve *BV*. While breakdown still occurs in the Si material, the breakdown voltage of GaN/Si VDMOS achieved a significant improvement, surpassing that of traditional Si VDMOS. TCAD simulation results demonstrate that the *BV* of GaN/Si VDMOS is 2029 V, and the *R_on,sp_* is 17.2 mΩ·cm^2^, effectively breaking the constraint relationship between *BV* and *R_on,sp_*. Therefore, a promising avenue is opened up for the development of vertical devices for future power applications.

## 2. Materials and Methods

Figure 1 depicts the cross-section of GaN/Si VDMOS. The preparation of GaN substrate provides a guarantee when achieving vertical GaN power devices [8]. Firstly, an N^-^type GaN layer is epitaxially grown on the N^+^ GaN substrate. Then, at room temperature (25 °C), GaN-Si can be directly wafer bonded through surface-activated bonding (SAB), a method that reduces surface energy through chemical bonding and achieves strong bonding at the atomic scale, thereby addressing the annealing and thermal stress issues caused by mismatched thermal expansion coefficients. This process facilitates the fabrication of GaN/Si heterogeneous junctions [27]. Subsequently, on the Si material, the P-well, N^+^ source, and P^+^ region are formed through ion implantation to enable the formation of ohmic contacts for the channel and source regions.

In this study, the two-dimensional numerical simulation of GaN/Si VDMOS was conducted using the ISE-TCAD simulation software. The main physics models include Mobility (DopingDep High Field Sat Enormal), EffectiveIntrinsic Density (OldSlotboom), Recombination (SRH (DopingDep), and Auger Avalanche (Eparal)). The criterion of breakdown is BreakCriteria {Current (Contact = “drain” Absval = 1 × 10^−7^)}. The breakdown condition was defined as the point at which the ionization integral equals unity. The parameter is Material = “GaN”. The coordinates make it necessary to optimize the parameters in the numerical simulations. Some of the device parameters in the simulation are presented in Table 1. The ambient temperature is 300 K, the breakdown voltage (*BV*) is obtained at *V_GS_* = 0 V, and the specific on-resistance (*R_on,sp_*) is obtained at *V_GS_* = 10 V, and the simulation results are as shown in Table 2.

## 3. Results and Discussion

The vertical electric field for GaN/Si VDMOS and conventional Si VDMOS is shown in Figure 2. In the case of Si VDMOS, when X = 2.5 μm, the maximum field strength reaches 2.55 × 10^5^ V/cm (reaching the critical breakdown electric field of Si materials), resulting in a *BV* of 374 V for Si VDMOS [19,20]. On the other hand, in GaN/Si VDMOS, when the drain voltage reaches 374 V, the electric field at the interface between the P base and the N-type drift region does not reach the critical breakdown field strength of GaN materials; therefore, the device does not breakdown. As the drain voltage continues to increase, the electric field strength of the device rises until the electric field at the heterojunction reaches 2.85 × 10^5^ V/cm (reaching the critical breakdown field of the Si materials), leading to the breakdown of GaN/Si VDMOS. Therefore, the *BV* of GaN/Si VDMOS increases from 374 V to 2029 V compared to the conventional Si VDMOS [21,22].

Figure 3a illustrates the blocking characteristics of GaN/Si VDMOS and Si VDMOS. Under the same bias conditions, the proposed GaN/Si VDMOS exhibits a significantly higher *BV* compared to conventional Si VDMOS. The results demonstrate that the *BV* of GaN/Si VDMOS is 2029 V, which is 443% higher than Si VDMOS. This improvement is attributed to the introduction of the GaN/Si heterojunction, which shifts the breakdown point of the new structure from a high-electric-field region (area A in Figure 1) to a low-electric-field region (area B in Figure 1), greatly increasing the breakdown voltage of the device. Figure 3b represents the output characteristics of GaN/Si VDMOS and Si VDMOS. When a sufficiently large positive bias voltage is applied to the gate, an n-channel inversion layer is formed in the p-well of the GaN/Si VDMOS, allowing for the current to flow from the drain to the source. In the case of GaN/Si VDMOS, the inversion channel forms between the N^+^ source region and the N^-^drift region beneath the gate oxide layer, on the Si side. The device’s resistance primarily depends on the channel resistance (*R_ch_*) and the JFET resistance (*R_JFET_*). The introduction of a GaN/Si heterojunction effectively adjusts the value of *R_ch_*, thereby influencing the overall resistance of the device. When the gate is positively biased, the GaN/Si VDMOS is in the on-state. The built-in electric field increases the electric field in the inversion layer, increasing the number of electrons in the channel and consequently increasing the electron current, resulting in a lower specific on-resistance of the device. The *R_on,sp_* of GaN/Si VDMOS is 17.2 mΩ·cm^2^, which is 53% lower than that of conventional Si VDMOS. Figure 3c displays the transfer characteristics curves of GaN/Si VDMOS and Si VDMOS. Typically, the threshold voltage (*V_TH_*) is determined by the gate oxide layer thickness and the surface concentration of Pwell. For a fixed gate oxide layer thickness, the *V_TH_* of GaN/Si VDMOS is 4.67 V, which is similar to Si VDMOS.

Figure 4a shows the energy band diagrams of GaN/Si VDMOS and Si VDMOS under thermal equilibrium. The bandgap of Si VDMOS is 1.12 eV, and the breakdown point is located at the junction between the P-based drift region and the N-type drift region, specifically at the maximum curvature radius of conventional Si VDMOS (X = 2.5 μm). GaN/Si VDMOS has a bandgap of 3.47 eV, and the breakdown point is located in the Si layer at the GaN/Si junction. The transfer of the breakdown point from one location to another is responsible for the increase in *BV* from 374 V to 2029 V in GaN/Si VDMOS. Figure 4b shows the band diagram of the N^+^-Si/P-GaN heterojunction (AA’), the P-Si/P-GaN heterojunction (BB’), the P-Si/N^−^-GaN heterojunction (CC’), and the N^−^-Si/N^−^-GaN in the middle of mesa (DD’). It is worth noting that the band diagram of the GaN/Si heterojunction in GaN/Si VDMOS is influenced by the doping concentrations and types in different regions of the device. The presence of the heterojunction in the channel and JFET region significantly affects the *R_ch_* in the device.

Figure 5 shows the simulated blocking characteristics, output characteristics and transfer characteristics of the conventional Si VDMOS and GaN/Si VDMOS with the different concentrations of interface state charge. The introduced interface state charge in this paper can be either donor (electron) or acceptor (hole). Compared to the conventional Si VDMOS (*BV* = 374 V, *R_on,sp_
*= 36.5 mΩ·cm^2^, *V_TH_
*= 5.55 V), the GaN/Si VDMOS without interface state charge exhibits an improved performance with a *BV* of 2029 V, *R_on,sp_* = 17.2 mΩ·cm^2^, and *V_TH_* = 4.67 V. The introduction of a p-type trap layer by acceptor interface charge induces an internal electron barrier in the inversion layer at the GaN/Si heterojunction. This enhances the internal electric field at the GaN/Si interface and changes the distribution of the electric field, resulting in an increase in the *BV* and *R_on,sp_*. As the concentrations of acceptor-like interface state charges increase from 1 × 10^11^ cm^−2^ to 5 × 10^11^ cm^−2^, the *BV* of GaN/Si VDMOS increases from 2057 V to 2308 V (shown in Figure 5a), the *R_on,sp_* increases from 17.7 mΩ·cm^2^ to 118.2 mΩ·cm^2^ for *L_D_
*= 20 μm (shown in Figure 5b), while the *V_TH_* of GaN/Si VDMOS decreases from 4.65 V to 4.64 V (shown in Figure 5c). The increase in acceptor-like interface state charges reduces the internal electric field at the GaN/Si heterojunction. Consequently, the internal electron barrier is induced in the inversion layer at the GaN/Si interface, leading to an increase in the electron channel current. The *R_on,sp_* drops from 16.8 mΩ·cm^2^ to 16.2 mΩ·cm^2^ due to the concentrations of interface state charges (donor) from 1 × 10^11^ cm^−2^ increasing to 5 × 10^11^ cm^−2^ (shown in Figure 5b), respectively; the *BV* are 2000 V and 1732 V (shown in Figure 5a), and the *V_TH_* of GaN/Si VDMOS increases from 4.69 V to 4.71 V (shown in Figure 5c).

The influence of different interface state charge (donor) on the energy band diagram of GaN/Si VDMOS is shown in Figure 6a. As the interface concentration increases, more electrons are induced in the inversion layer at the GaN/Si heterojunction interface. This leads to an increase in the number of electrons in the channel, resulting in a decrease in *R_on,sp_*. Additionally, the decrease in the barrier height causes a reduction in the *BV* of the GaN/Si VDMOS. Figure 6b shows the influence of different interface state charges (acceptorlike) on the energy band diagram of GaN/Si VDMOS. The introduction of acceptor interface charge at the GaN/Si heterojunction interface increases the barrier height difference at the heterojunction. Consequently, the barrier height becomes higher as the interface concentration increases. Further, the *BV* and the *R_on,sp_* of the GaN/Si VDMOS increase (shown in Figure 3a,b).

The influence of different interface state charges concentrations on the characteristics of the GaN/Si VDMOS is shown in Figure 7. The device is positively biased, with the gate being forward-biased, causing changes in the internal electric field at the GaN/Si heterojunction as the concentration and type of interface state charges vary. Compared with GaN/Si VDMOS without an interface state charge, the *BV* increased due to an elevation in the concentration of acceptor-like interface state charges. This is because the p-type trap layer is introduced by the interfacial charge (acceptorlike) at the GaN/Si heterojunction, resulting in the increase in the *BV*. When the concentrations of acceptor-like interface state charges increase from 1 × 10^11^ cm^−2^ to 5 × 10^11^ cm^−2^, the *BV* of GaN/Si VDMOS increases from 2057 V to 2308 V, and the *R_on,sp_* increases from 17.7 mΩ·cm^2^ to 118.2 mΩ·cm^2^ for *L_D_
*= 20 μm. On the other hand, an increase in the concentration of donor-like interface state charges results in a decrease in the internal electric field at the GaN/Si heterojunction. This induces an internal electron barrier in the inversion layer at the GaN/Si interface and increases the number of electrons in the channel, thereby augmenting the electron current. This results in a lower *R_on,sp_* when the concentrations of interface state charges are 0, 1 × 10^11^ cm^−2^, 3 × 10^11^ cm^−2^ and 5 × 10^11^ cm^−2^, respectively; the *R_on,sp_* drops from 17.2 mΩ·cm^2^ to 16.2 mΩ·cm^2^, and the *BV* is 2029 V, 2000 V, 1860 V, and 1732 V.

The influence of different concentrations of acceptor-like interface state charges on *D_Si_* for GaN/Si VDMOS is illustrated in Figure 8a, while the effect of different concentrations of donor-like interface state charges on *D_Si_* for GaN/Si VDMOS is shown in Figure 8b. The internal electric field at the GaN/Si heterojunction varies with the concentration and type of interface state charge. When the interface state type is acceptor-like, a p-type trap layer is introduced at the GaN/Si heterojunction, altering the built-in electric field at the heterojunction and increasing the breakdown voltage of the device. For *D_Si_* = 0.5 μm, as the interface state concentration increases, the *BV* of GaN/Si VDMOS rises from 2057 V to 2308 V. Simultaneously, due to the presence of the p-type trap layer, the electron concentration in the inversion channel is significantly reduced, leading to an increase in the *R_on,sp_* of the device. As the interface state concentration increases, the *Ron,sp* of GaN/Si VDMOS increases from 17.7 mΩ·cm^2^ to 118.2 mΩ·cm^2^. It can be observed that a decrease in *D_Si_* significantly improves the *BV*. This is because a decrease in *D_Si_* implies an increased distance between the heterojunction and area A, thereby increasing the electric field difference between area A and area B (as shown in Figure 1). Consequently, the breakdown point shifts to a lower-electric-field region, with lower electric field in area B and higher electric field in area A, resulting in a further increase in the *BV* of GaN/Si VDMOS. Meanwhile, as *D_Si_* increases, the influence of the built-in electric field on the electric field in the inversion layer decreases, leading to an increase in the *R_on,sp_*.

Figure 9 shows the dependences of *BV* and *R_on,sp_* on *D_Si_* and *L_D_* for GaN/Si VDMOS with the concentrations of interface state charge of −3 × 10^11^cm^−2^ and 3 × 10^11^cm^−2^. The *D_Si_* includes the thickness of Si for the GaN/Si VDMOS. As the *D_Si_* decreases, the *BV* improves. This improvement is attributed to the increasing electric field difference between area A and area B, which is a result of the rising interface of the GaN/Si heterojunction and the continuing interface effect. A lower electric field is found at area B compared to area A, the higher the *BV* in the GaN/Si VDMOS is; meanwhile, the *R_on,sp_* increases slightly. This study considers interface state charges of donor (electron) or acceptor (hole) type, with concentrations ranging from 1 × 10^11^cm^−2^ to 5 × 10^11^cm^−2^. Therefore, the concentrations of acceptor-like interface state charge of 3 × 10^11^cm^−2^ (shown in Figure 9a) and the concentrations of donor-like interface state charges of 3 × 10^11^cm^−2^ (shown in Figure 9b) are considered in this case. The *BV* (*BV* = 2184 V) can be significantly increased with the same *L_D_* of 20 μm, and the *R_on,sp_* (*R_on,sp_
*= 25.83 mΩ·cm^2^) also decreased with the *D_Si_* of 0.5 μm, compared with the *D_Si_* of 2 μm for GaN/Si VDMOS (*BV* = 975 V, *R_on,sp_
*= 41.79 mΩ·cm^2^) (shown in Figure 9a). Similarly, with the same *D_Si_* of 0.5 μm and an *L_D_* of 20 μm, the *R_on,sp_* (*R_on,sp_* = 16.5 mΩ·cm^2^) significantly increases, and the *BV* (*BV* = 1860 V) increases with the *L_D_* of 20 μm, compared with the *L_D_* of 5 μm for GaN/Si VDMOS (*BV* = 725 V, *R_on,sp_
*= 6.08 mΩ·cm^2^) (shown in Figure 9b).

The dependences of *BV*, *R_on,sp_* and Figure-Of-Merit (FOM = *BV*^2^/*R_on,sp_*) on *L_D_* for the conventional Si VDMOS and GaN/Si VDMOS with the concentrations of interface state charge of −3 × 10^11^ cm^−2^ and 3 × 10^11^ cm^−2^ are shown in Figure 10. The *BV* of GaN/Si VDMOS exhibits a faster increase as the *L_D_* increases (*BV* = 2184 V, at *L_D_
*= 20 μm and the concentrations of interface state charges (acceptorlike) of 3 × 10^11^ cm^−2^) compared with the conventional Si VDMOS (*BV* = 374 V, at *L_D_
*= 20 μm). Furthermore, the *R_on,sp_* of GaN/Si VDMOS (*R_on,sp_
*= 25.83 mΩ·cm^2^, at *L_D_
*= 20 μm) is lower than that of the conventional Si VDMOS (*R_on,sp_
*= 36.48 mΩ·cm^2^, at *L_D_
*= 20 μm), yielding to the higher FOM (184.7 MW/cm^2^) for GaN/Si VDMOS than that (3.8 MW/cm^2^) of the conventional Si VDMOS with the same *L_D_* of 20 μm. Additionally, the FOM (209.6 MW/cm^2^) and *BV* (*BV* = 1860 V, at *L_D_
*= 20 μm and the concentrations of interface state charges (donor) of 3 × 10^11^ cm^−2^) of GaN/Si VDMOS demonstrate an improvement, and *R_on,sp_
*(16.5 mΩ·cm^2^) is reduced compared with the conventional Si VDMOS (shown in Figure 10).

Figure 11 shows the simulated output and transfer characteristics of the conventional Si VDMOS and GaN/Si VDMOS with the concentrations of an interface state charge of 3 × 10^11^ cm^−2^. The threshold voltage *V_TH_* of 6.72 V for GaN/Si VDMOS is higher than that of the conventional Si VDMOS (*V_TH_
*= 5.63 V). At the different gate voltages *V_GS_* (5.5, 6, 6.5, 10 V), GaN/Si VDMOS exhibits a better output performance than traditional Si VDMOS, resulting in a lower *R_on,sp_
*(16.52 mΩ·cm^2^) of GaN/Si VDMOS compared to traditional Si VDMOS (35.65 mΩ·cm^2^).

The *R_on,sp_* versus *BV* in GaN/Si VDMOS and other reported VDMOS devices are compared in Figure 12 [8,9,10,11,12,13,14,15,16,17]. It can be observed that GaN/Si VDMOS exhibits a *BV* of 2029 V and a *R_on,sp_* of 17.2 mΩ·cm^2^, resulting in an excellent Baliga figure-of-merit (FOM) of 239.4 MW/cm^2^. These results indicate that the proposed novel structure surpasses the performance limitations of silicon.

## 4. Conclusions

The novel GaN/Si VDMOS with the GaN/Si heterojunction is proposed in this letter, based on the advantages of GaN materials. The novel GaN/Si VDMOS can not only increase the *BV* (*BV* = 2029 V) but also reduces the *R_on,sp_* (*R_on,sp_
*= 17.2 mΩ·cm^2^) compared with the traditional Si VDMOS (*BV* = 374 V, *R_on,sp_
*= 36.48 mΩ·cm^2^) because the breakdown point is transferred from the higher-electric-field region with the maximum curvature radius to the low-electric-field region by BPT. The interfacial effect of the GaN/Si heterojunction greatly affects the characteristics of the device due to the introduction of the GaN/Si heterojunction.

## Figures and Tables

**Figure 1 micromachines-14-01166-f001:**
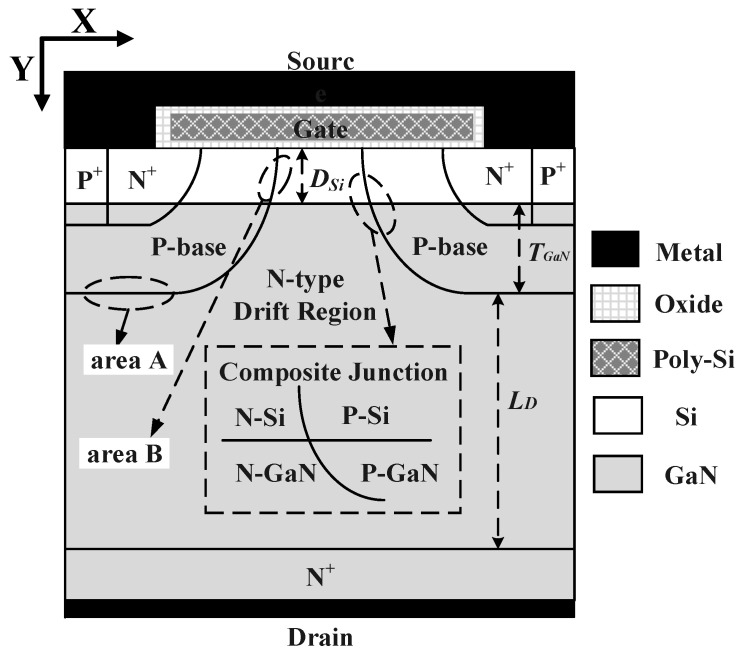
Cross-section of the novel GaN/Si VDMOS.

**Figure 2 micromachines-14-01166-f002:**
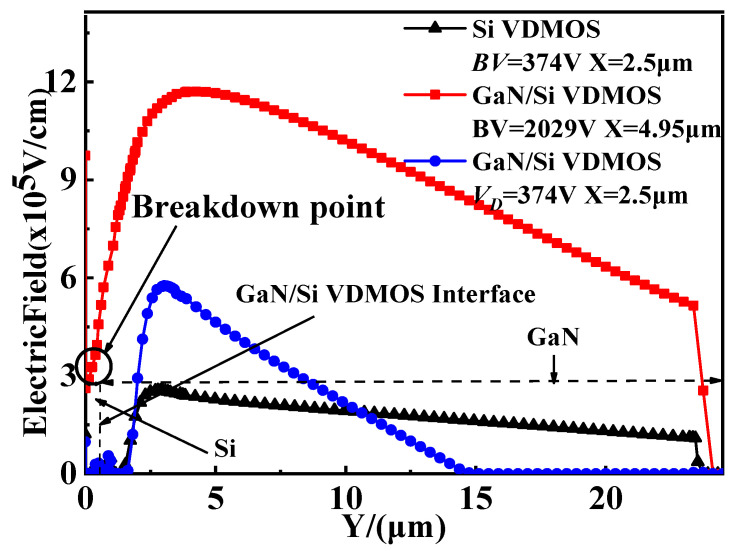
Vertical electric field distributions for GaN/Si VDMOS and Si VDMOS: *L_D_
*= 20 μm, *D_Si_* = 0.5 μm.

**Figure 3 micromachines-14-01166-f003:**
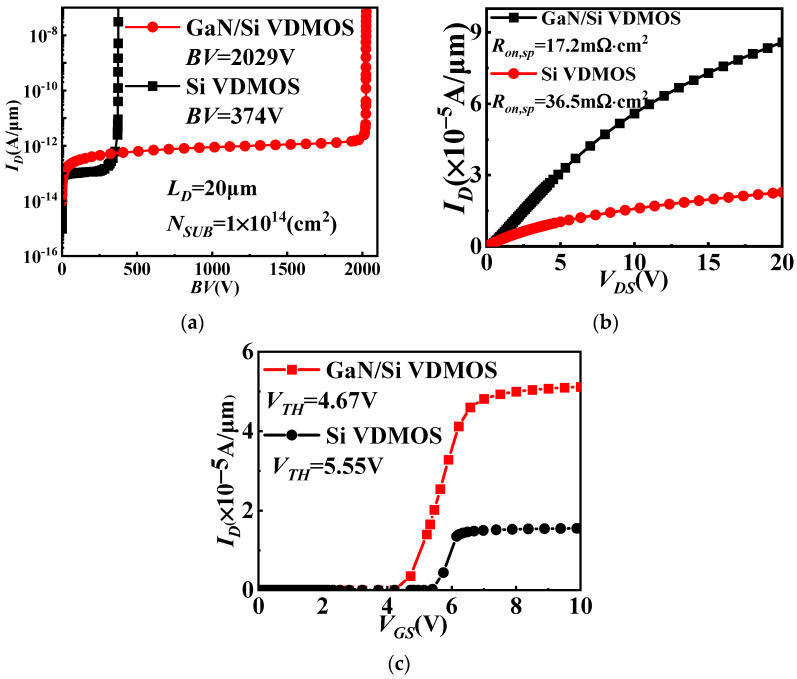
(**a**) Blocking characteristic, (**b**) output characteristics and (**c**) transfer characteristics for GaN/Si VDMOS and Si VDMOS.

**Figure 4 micromachines-14-01166-f004:**
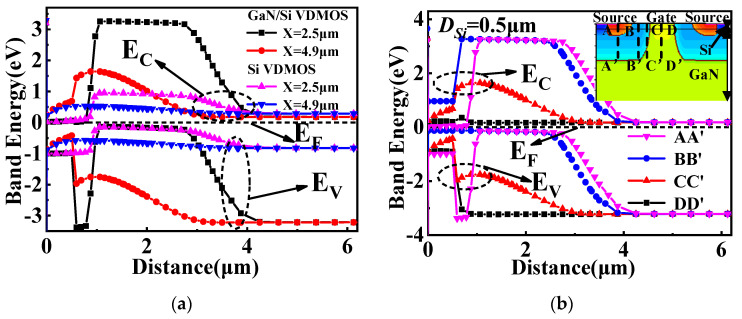
(**a**) The energy band diagram at thermal equilibrium for the GaN/Si VDMOS and the Si VDMOS, and (**b**) Band diagram of the N^+^-Si/P-GaN heterojunction (AA’), the P-Si/P-GaN heterojunction (BB’), the P-Si/N^−^-GaN heterojunction (CC’), and the N^−^-Si/N^−^-GaN in the middle of mesa (DD’).

**Figure 5 micromachines-14-01166-f005:**
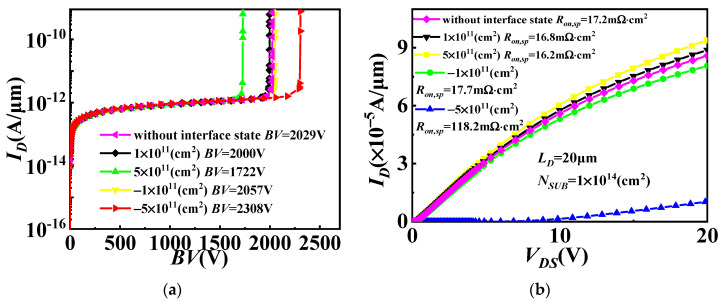
(**a**) Blocking characteristic, (**b**) output characteristics and (**c**) transfer characteristics for GaN/Si VDMOS with the different concentrations of interface state charge.

**Figure 6 micromachines-14-01166-f006:**
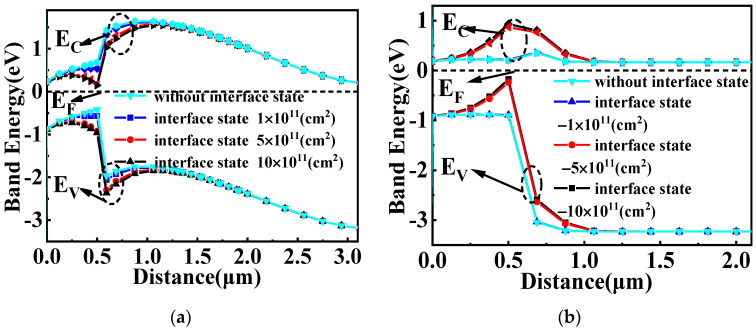
(**a**) The influence of different interface state charge (donor) on the energy band diagram of GaN/Si VDMOS and (**b**) the influence of different interface state charge (acceptorlike) on the energy band diagram of GaN/Si VDMOS: *L_D_
*= 20 μm, *D_Si_* = 0.5 μm.

**Figure 7 micromachines-14-01166-f007:**
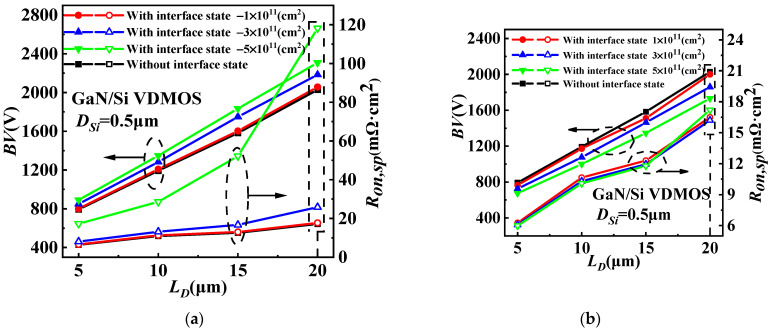
(**a**) The influence of different interface state charges (acceptor-like) on the characteristics of GaN/Si VDMOS on *L_D_* and (**b**) the influence of different interface state charges (donor) on the characteristics of GaN/Si VDMOS on *L_D_*.

**Figure 8 micromachines-14-01166-f008:**
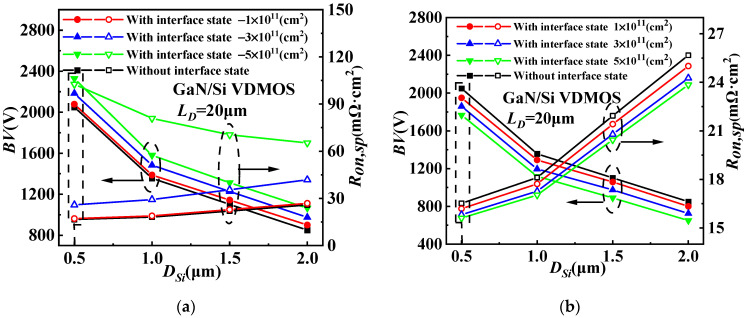
(**a**) The influence of different concentrations of acceptor-like interface state charges on *D_Si_* for GaN/Si VDMOS and (**b**) the influence of different concentrations of donor-like interface state charges on *D_Si_* for GaN/Si VDMOS.

**Figure 9 micromachines-14-01166-f009:**
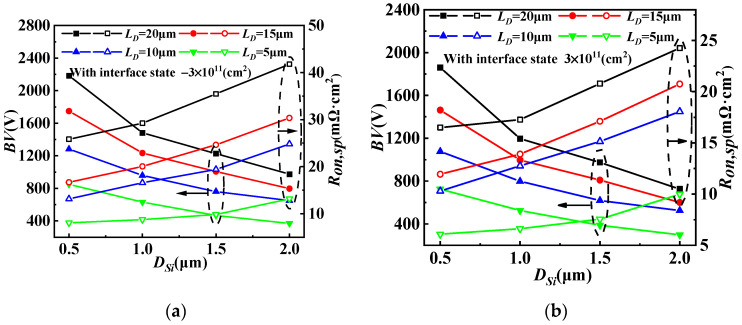
Dependences of *BV* and *R_on,sp_* on *D_Si_* and *L_D_* for GaN/Si VDMOS with (**a**) the concentrations of interface state charge (acceptorlike) of 3 × 10^11^ cm^−2^ and (**b**) with the concentrations of interface state charges (donor) of 3 × 10^11^ cm^−2^.

**Figure 10 micromachines-14-01166-f010:**
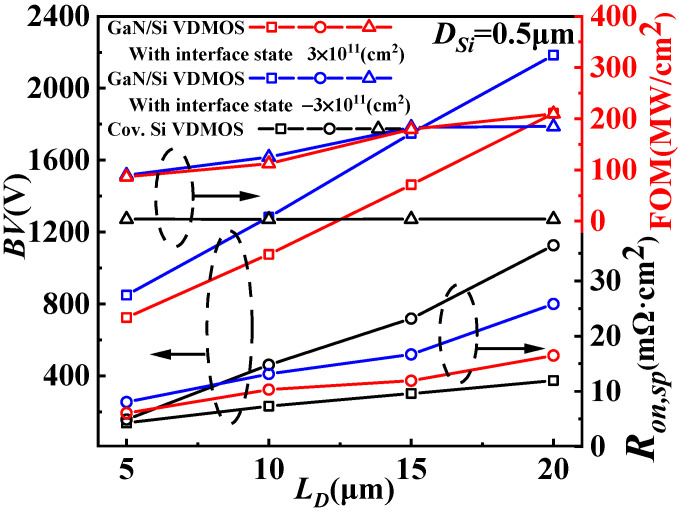
Dependences of *BV*, *R_on,sp_* and figure-of-merit (FOM) on *L_D_* for the conventional Si VDMOS and GaN/Si VDMOS with the concentrations of an interface state charge of −3 × 10^11^ cm^−2^ and 3 × 10^11^ cm^−2^.

**Figure 11 micromachines-14-01166-f011:**
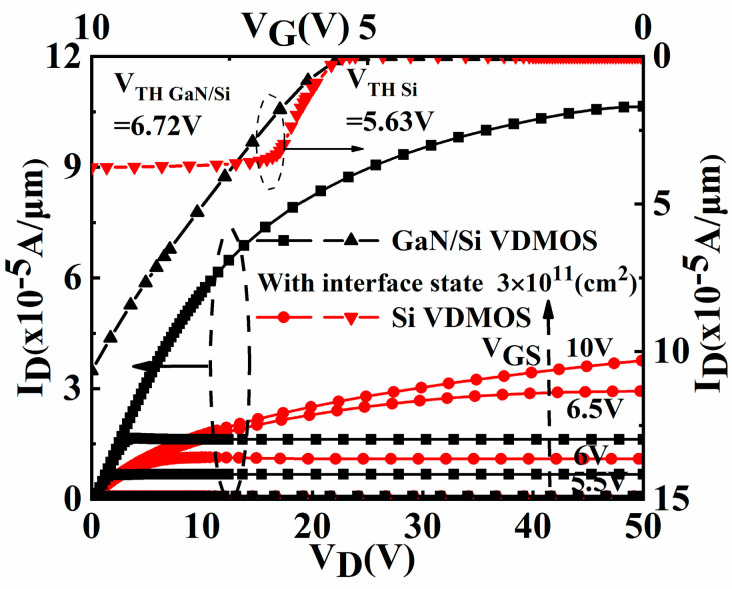
Output characteristics and transfer characteristics of the conventional Si VDMOS and GaN/Si VDMOS with the concentrations of interface state charge of 3 × 10^11^ cm^−2^.

**Figure 12 micromachines-14-01166-f012:**
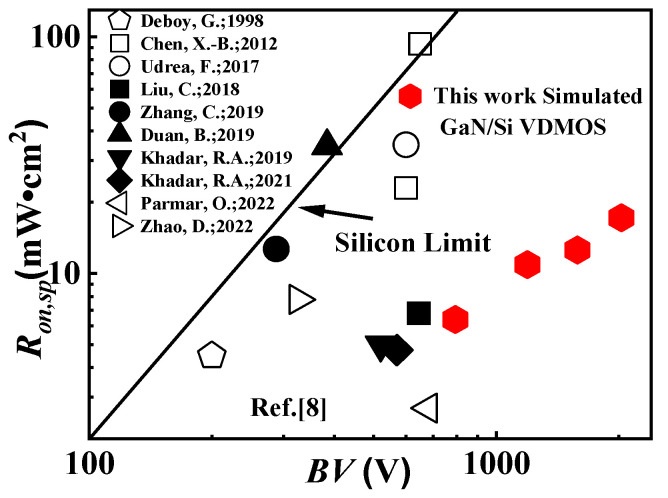
The *R_on,sp_* versus BV with the ideal silicon limit line [8,9,10,11,12,13,14,15,16,17].

**Table 1 micromachines-14-01166-t001:** Device parameters in the simulation.

Device	Si VDMOS	GaN/Si VDMOS
*D_si_ *(μm)	/	0.5
*T_GaN_ *(μm)	/	3
*L_D_ *(μm)	20	20
*N_D_ *(cm^−3^)	3.5 × 10^14^	2.4 × 10^15^
*N_p_ *(cm^−3^)	5 × 10^17^	5 × 10^17^
*N_SUB_ *(cm^−3^)	1 × 10^14^	1 × 10^14^

*D_Si_* is the thickness of Si for the GaN/Si VDMOS. *T_GaN_* is the depth of the GaN at the P-base region. *L_D_* is the length of N^-^drift region. *N_D_* is the concentration of N^-^drift region. *N_SUB_* is the concentration of P-substrate. *N_P_* is the concentration of P-base region.

**Table 2 micromachines-14-01166-t002:** Simulation results for the Si VDMOS and GaN/Si VDMOS.

Device	Si VDMOS	GaN/Si VDMOS
*BV* (V)	374	2029
*R*_on,sp_ (Mω·cm^2^)	36.5	17.2
*FOM* (MW/cm^2^)	3.83	239.4

## Data Availability

Not applicable.

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
