# Peer review of "GaN/Si Heterojunction VDMOS with High Breakdown Voltage and Low Specific On-Resistance"

_micromachines, 2023, doi:10.3390/mi14061166_

Round 1
Reviewer 1 Report
In this manuscript, GaN/Si heterojunction (GaN/Si VDMOS) is proposed to optimize the breakdown voltage (BV) and the specific on-resistance (Ron,sp). GaN material as a substrate heteroepitaxial Si material, breaking the limit of Si-based power semiconductor devices. Using wide bandgap GaN materials as the substrate and drift region of the device has effectively improved device performance. The new structure combining the “soft” material of Si with the “hard” material of GaN playing their advantages. Due to the active region of the device is formed in the Si material, the mature silicon process can be utilized in device manufacturing to achieve better ohmic contact. I consider the efforts of the authors to be interesting and meaningful. However, there are some questions and confusion in the manuscript need to be solved before publication.
1. The breakdown criterion is considered in this paper is 10-7A, which is comparable to the ON current of the existing devices and proposed devices. The off current is too large for the purpose of breakdown. An appropriate limit of breakdown current shall be used to find the BDV.
2. For the breakdown model, are both electrons and holes counted for the BDV and, current at BDV estimation? Please complement the model used in this paper.
3. Some sentences need to be added of how to produce such GaN/Si heterojunction are necessary.
4. The power MOS structures were significantly improved over the years and many improvements can be found in literature. The authors should add a comparison with existing structures in the literature.
The English grammar and usage are in need of improvement.
Reviewer 2 Report
The work submitted by Yang et al presents simulation studies about a possible novel structure: GaN/si heterostructure, which could provide a better performance in the field of power electronics related devices.
The topic could be, in principle, interesting for this specific field. However, the manuscript, in the present form, is not ready to be accepted in the journal MIcromachines.
The introduction should be amended, where the following aspects are discussed in more depth: the novelty of the work, its motivation and how the device proposed by the authors and subject of the simulations outperforms the similar devices that are the most used or/and studied currently, at the commercial level and in comparison with other recent works in the field. Therefore, this also includes the addition of more recent and state-of-the art articles in the section of references.
How the parameters used for the simulations and presented in table 1 were chosen, why and how they are comparable with real devices, as well as a more detailed description of the simulation method and tools developed here, are missing.
The results and discussion section is mainly a description of the observed results, a description of the values shown through the graphs presented in figures 2 to 11. Please, I kindly ask to provide a more critical and detailed analysis of the results and compare them with the results obtained from other similar devices, both simulated and fabricated experimental ones.
Some sentences are difficult to understand. However, since the manuscript requires a heavy amount of amendment, I would comment with more detail about the quality of English language after the resubmission of an improved version of the work.
Reviewer 3 Report
According to the authors, GaN/Si-based VDMOS has a high breakdown voltage and specific Ron characteristics compared to other VDMOS. Despite this, the discussion lacks consistency. In its current form, the manuscript does not indicate how the work is novel or original. The physics behind their results is difficult to comprehend, especially in Figure 2. How do the authors define the breakdown point? The authors consider interface charges of ~1011 cm-2. There is no reference to that consideration. Why is specific Ron very low for GaN/Si-based VDMOS? Detail explanation is required. Is it possible to grow Si on GaN directly in terms of an actual device? Figure 4 should be represented.
Checking English should be done by native speakers.
Reviewer 4 Report
In my judgement, according to the revisions made by authors, the work can be published on Micromachines.
In my judgement, according to the revisions made by authors, the work can be published on Micromachines.
Round 2
Reviewer 2 Report
Thank you very much for replying all my comments point by point.
The manuscript has improved significantly and it is ready to be accepted in Micromachines.
In general, a significant amount of sentences are quite long dealing with English. An example is: "However, due to the technical immaturity of Si-based GaN devices in terms of source ohmic contacts and gate oxidation processes, their manufacturing poses significant challenges, resulting in difficulties in achieving optimal electrode formation, thereby greatly limiting the development and application of GaN devices [8-9]."
Reviewer 3 Report
Still the authors needs improved the explanation their results.
Manuscript should be checked by the native speakers.